

# Unbalanced relationships: insights into the interaction between gut microbiota, geohelminths, and schistosomiasis

Matheus Pereira de Araújo[1,2,*], Marcello Otake Sato[2,*], Megumi Sato[3], Kasun M. Bandara WM[4], Luiz Felipe Leomil Coelho[1], Raquel Lopes Martins Souza[1], Satoru Kawai[2] and Marcos José Marques[1]

[1] Institute of Biomedical Sciences, Universidade Federal de Alfenas, Alfenas, Minas Gerais, Brazil
[2] Laboratory of Tropical Medicine and Parasitology, Dokkyo Medical University, Mibu, Tochigi, Japan
[3] Graduate School of Health Sciences, Niigata University, Niigata, Niigata, Japan
[4] The Open University Sri Lanka, Nawala, Nugegoda, Sri Lanka
[*] These authors contributed equally to this work.

## ABSTRACT

Hosts and their microbiota and parasites have co-evolved in an adaptative relationship since ancient times. The interaction between parasites and intestinal bacteria in terms of the hosts' health is currently a subject of great research interest. Therapeutic interventions can include manipulations of the structure of the intestinal microbiota, which have immunological interactions important for modulating the host's immune system and for reducing inflammation. Most helminths are intestinal parasites; the intestinal environment provides complex interactions with other microorganisms in which internal and external factors can influence the composition of the intestinal microbiota. Moreover, helminths and intestinal microorganisms can modulate the host's immune system either beneficially or harmfully. The immune response can be reduced due to co-infection, and bacteria from the intestinal microbiota can translocate to other organs. In this way, the treatment can be compromised, which, together with drug resistance by the parasites makes healing even more difficult. Thus, this work aimed to understand interactions between the microbiota and parasitic diseases caused by the most important geohelminths and schistosomiasis and the consequences of these associations.

## INTRODUCTION

The microbiota is important for humans because it is involved in many of the host's physiological processes, including the acquisition of nutrients and the development of the immune system. The term "microbiota", which refers to microorganisms present at a particular site in an organism, is determined by its diversity and number of species present, the activity it exerts on the organism, and the relationship with the host; there may be synergism or even competition of these species for the habitat (*Turnbaugh et al., 2007*). However, organisms other than the commensals can interfere with host homeostasis.

Corresponding authors
Marcello Otake Sato,
marcello@dokkyomed.ac.jp
Megumi Sato,
satomeg@clg.niigata-u.ac.jp

Parasites, such as helminths, can colonize the same environment, and once they are together with bacteria, they may lead to imbalances and even obstruction of the gut. This can lead to changes in the absorption of nutrients and result in severe malnutrition. Parasitic diseases affect millions of people around the world, mainly in countries undergoing industrialization (*Stensvold & Van der Giezen, 2018*). It is estimated that almost a quarter of the world's population is infected with STH, with many patients presenting multiple infections, demanding urgent care. Since parasites can interact with the host microbiota, the composition of intestinal bacteria can be a tool to modulate the immune system with the progression of parasites to prevent intestinal infection (*Turnbaugh et al., 2007*; *World Health Organization, 2020*).

The gut microbiota can be altered by factors such as new treatment modalities, immunization, and sanitation tactics (*Turnbaugh et al., 2007*; *Stensvold & Van der Giezen, 2018*). Advances in medical technology and health systems and changes in the population's lifestyle have been directly reflected in the treatment of parasitic diseases (*Moser, Schindler & Keiser, 2017*). However, studies related to neglected tropical diseases (NTDs) are poorly funded in developed countries, which has led to a shortage of new drugs. Due to the deficit in attention to infectious diseases, few alternatives are being developed for treatment in cases of parasites' resistance to drugs (*Idris, Wintola & Afolayan, 2019*). Alterations in the gut microbiota can be directly associated with the permeability of intestinal mucosa, inflammatory disorders, and immune dysregulation, leading to autoimmune disorders (*Ajslev et al., 2011*). Since many parasites can interact with the host microbiota, challenges of new treatments in individuals with these parasites can be observed, with consequences in the development of a disease (*Hooper, Littman & Macpherson, 2012*; *Glendinning et al., 2014*). Helminths have a detrimental effect on the host. Foodborne and waterborne parasitic diseases are important worldwide, and they result in millions of deaths every year (*World Health Organization, 2019*). In endemic areas, where infected individuals excrete eggs and larvae, there may be contamination of the soil and food; thus, also considering the lack of government actions to avoid the transmission, the proliferation of these parasites is more frequent (*Idris, Wintola & Afolayan, 2019*). Helminth diseases around the world, including ascariasis, trichuriasis, ancylostomiasis, and schistosomiasis, can have direct interaction with the gut microbiota (*World Health Organization, 2020*).

Ascariasis is the most common helminthic infection worldwide, with an estimated more than a billion infected individuals (*Hailegebriel, Nibret & Munshea, 2020*). It has been reported that the disease-causing species can reduce the diversity of the gut microbiota (*Wang et al., 1999*). Similarly, trichuriasis can not only change the composition of the gut microbiota but also the bacterial invasion of the large intestinal epithelium of infected mice; however, in humans, it does not change the microbiota even after treatment (*Schachter et al., 2020*; *Cooper et al., 2013*). Unlike ascariasis and trichuriasis, hookworm infection in humans can increase gut bacterial diversity, suggesting a possible role in hookworm-induced enteritis. Correspondingly, the alteration of gut microbiota in humans and mice infected with *Schistosoma* spp. has been well elucidated (*Schneeberger et al., 2018*; *Cortés et al., 2020*; *Gordon et al., 2020*; *Hu et al., 2020*). Thus, the present article reviews the current

knowledge about the interactions between geohelminths/schistosomiasis with the host gut microbiota and their significance in health alterations.

## SURVEY METHODOLOGY

This was an integrative review with data collection carried out from sources through a bibliographic survey. To ensure an unbiased review of the literature, we performed a search of the following databases: MedLine, Web of Science, Scielo, and PubMed.

The following descriptors and their combinations in English were used to search for articles: ''Helminths Microbiota'' and ''Parasite Microbiota'' in combination with ''Ascariasis'', ''Trichuriasis'', ''Ancylostomiasis'', and ''Schistosomiasis'', along with using ''+'', ''AND'', and ''OR'' for a specific search result. The identified papers were initially checked to determine their appropriateness to the subject, and all of the relevant articles were read in detail. We also examined relevant papers referenced and identified during the initial search.

The inclusion criteria defined for the selection of the articles were as follows: primarily articles published in English, along with Portuguese or Spanish; full-text articles that portrayed the theme related to the association with parasites and how it alters the microbiota of the host, with focus on helminths; and articles that had been published in the last 15 years. The analysis was performed by gathering the data extracted from the articles in a descriptive way, making it possible to describe, observe, and classify the data, to synthesize the knowledge on the topic chosen in this review. Grey literature and papers that did not meet the inclusion criteria were excluded.

## GASTROINTESTINAL MICROBIOTA AND THEIR INTERACTION WITH THE HOST

The studies of gastrointestinal (GI) microbiota involved microbial diversity, an abundance of species present, their activity, and their competitive interaction and synergism. The interaction between the gastrointestinal tract and the resident microbiota is well balanced in healthy individuals, but in disequilibrium, it can lead to diseases (Table 1) (*Jenkins et al., 2021*). The misuse of antibiotics, dietary changes, and other infections, such as helminths that compete for the same habitat, have received growing attention with regard to pathogen–host interaction and imbalances in GI microbiota that favor opportunistic infections (*Zoetendal et al., 2004*).

Most of the bacteria present in the human gastrointestinal tract are not harmful but rather beneficial. Microbial profiles and their concentration, including several communities of commensal microorganisms, vary depending on the different habitats in the gastrointestinal tract, which present different pH levels and oxygen concentrations (*Foulongne et al., 2012*; *Belkaid & Harrison, 2017*). Oral microbiota is a heterogeneous ecological system that protects from the colonization of bacteria that could affect systemic health. The buffering capacity of saliva is well recognized as a major factor that influences the configuration of oral microbiota in humans. Saliva further facilitates the formation of acquired pellicles on the surface of the oral cavity, which leads to initial adhesion, colonization, and makeup of

Pereira de Araújo et al. (2022), *PeerJ*, DOI 10.7717/peerj.13401

**Table 1  Interaction of helminths with human host microbiota and their effects.**

| Helminth | Host microbiota interaction | Immune response change | Effects to the human host | References |
|---|---|---|---|---|
| *Ascaris* sp. | Can modulate the human gut microbiota | Immune system and metabolic activities are influenced by nematode's Excretory-Secretory products | Induction of human depression symptomatically | *Acevedo et al. (2011)*, *Midha et al. (2018)* *Ramírez-Carrillo et al. (2020)* |
| *Trichuris* sp. | Can modulate mouse intestinal microflora | May influence the immune system of children immensely and reduce the allergen skin test reactivity significantly | Regulation of chronic responses of the parasite infected and potential lasting immunological tolerance of intestinal dysbiosis reduce the development of asthmatic reactions in their later childhood | *Elliott, Mpairwe & Quigley (2005)*, *Rodrigues et al. (2008)* *Houlden et al. (2015)* |
| *Necator americanus/ Ancylostoma duodenale* | Appear not influence depletion of microbiota diversity | Could be related to allergic or immunological disarrangement alleviation | Low-dose of hookworm administration is being used as therapeutic interventions for certain human chronic diseases | *Hooper, Littman & Macpherson (2012)*; *Loukas et al. (2016)*; *Hailegebriel, Nibret & Munshea (2020)* |
| *Schistosoma* sp. | Anthelmintic changes the composition of the microbiota irreversibly | Lesions in the intestinal epithelium which, associated with the immunomodulatory response, can lead to a decrease in the protective barrier against bacteria | Despite antibiotics and anti-mycotics reduce both gut microbiota and inflammation significantly, resulting a less granuloma development, the association of bacteria with *S. mansoni* enables prolonged bacterial infections, the development of antibiotics resistance, and the ineffective treatment of both infections. | *Muniz-Junqueira, Tosta & Prata (2009)*; *Barnhill et al. (2011)*; *Holzscheiter et al. (2014)* |

the resident bacteria. This is the medium that delivers nutrients and trace elements such as glycoproteins, albumins, acidic proline, sialic acids, and mucins for bacterial survival and growth (*Cornejo Ulloa, Van der Veen & Krom, 2019*).

Recent culture-independent studies have revealed that the esophagus contains diverse microorganisms, which are mainly divided into two types. Type I is dominated by the genus *Streptococcus*, which is involved in dysplasia and inflammatory foci. Genera *Prevotella, Actinomyces, Lactobacillus,* and *Staphylococcus* have been reported as esophageal bacteria. Some bacterial groups, such as *Streptococci,* may include strains that extend their habitats from the oral cavity to the esophageal mucosa and the stomach in the absence of *Helicobacter pylori* infection (*Yang et al., 2009*; *Sekirov et al., 2010*).

Although the composition of the gastric microbiota is relatively poor due to the low pH, the stomach holds a diverse microbiota dominated by *Rothia, Streptococcus, Veillonella,* and *Prevotella*, when *H. pylori* is low in abundance or absent (*Sekirov et al., 2010*). In contrast, there is a shift in the abundance of *Streptococcus, Prevotella,* and Firmicutes phylum in *H. pylori*-infected stomach and gastric cancer (*Nardone & Compare, 2015*). The role of *H. pylori* in the development of peptic ulcers, gastritis, and adenocarcinoma is well defined. A previous study found 128 bacterial phylotypes and suggested a much more diverse gastric ecosystem than earlier described (*Bik et al., 2006*). In the human small intestine, the abundance of bacterial community increases in the proximal to distal direction. Based on early molecular assays, the genus *Streptococcus* appears to be the most common genus in the duodenum and jejunum (*Hollister, Gao & Versalovic, 2014*). Most densely populated and diversified microbiota is present in the lower part of the intestine, and is mainly dominated by phyla Firmicutes and Bacteroidetes, followed distantly by Verrucomicrobia and Actinobacteria (*Andersson et al., 2008*). The phylum Firmicutes in the human and animal intestinal microbiota comprises several clinically important genera such as *Staphylococcus,* lactic acid bacteria (LAB), and *Listeria spp.* (*Lanza et al., 2015*).

The vertebrate intestinal microbiota influences the development and balance of the immune system, and it has been studied in the prevention of damage induced by opportunistic bacteria as well as in the influence of systemic autoimmune diseases (*Ogaki, Furlaneto & Maia, 2015*). Another study has shown that cells of the immune system acquire distinct functional properties in response to intestinal commensals and pathogenic microbiota. Signals to modulate the innate immune system are conveyed by intestinal bacteria as a result of stimulation of innate immune "pattern recognition receptors (PRRs)" (*Ivanov et al., 2009*). The intestinal microbiota is also involved in the priming and maturation of the adaptive immune system (*Ramirez et al., 2020*), but it is poorly understood how these individual bacteria determine the location and type of immune response (*Ivanov et al., 2009*).

Recent advances in culture-independent methods to study microbes have suggested that antibiotics treatment adversely affects the intestinal microbiota, including the selection of antibiotic-resistant organisms, alteration of metabolic activity, and reduction of bacterial diversity, thereby resulting in short- and long-term health consequences such as gastrointestinal infections, obesity, colorectal cancer, and inflammatory bowel disease (IBD) (*Ramirez et al., 2020*). Broad-spectrum antibiotics are a major predisposing

factor for recurrent *Clostridium difficile* infections, which in turn can lead to antibiotic-associated diarrhea (*Stoddart & Wilcox, 2002*). Antibiotics are used to treat *H. pylori* infection, which produces an inflammatory response in the gastric mucosa. Nevertheless, *H. pylori* eradication by antibiotics can have both positive and negative impacts on the host's health. Rapid intestinal colonization by *Escherichia coli* (*E. coli*) EMO plays a pivotal role in protecting against enteropathogenic agents such as *Shigella flexneri* strains or *Salmonella enteritidis* subsp. *typhimurium.* Coadministration of probiotic agents such as *Lactobacillus acidophilus*, *Saccharomyces boulardii,* and *Escherichia coli* EMO to germ-free mice progressively improved their health (*Filho-Lima, Vieira & Nicoli, 2000*; *Hudault, Guignot & Servin, 2001*). These findings showed the beneficial role of the commensal microbiota in protecting against infection by pathogens in germ-free mice, concluding that those with an already established microbiota have a competent immunological defense system. However, the total health effects after manipulation of the composition of microbiota at each site of the gastrointestinal tract remain to be elucidated (*Coman & Vodnar, 2020*). Inappropriate host response due to complex intestinal commensal microbial community and their alteration is defended by joint action of epithelial cells, released mucus, and immunoglobulins in the intestinal mucosal barrier, further preventing IBDs. This protection is necessary to maintain homeostasis because the host's microbiota tries to minimize contact with invading microorganisms, reduces tissue inflammation, and prevents a possible translocation of microorganisms to other sites of the intestine (*McGuckin et al., 2009*). It is also important to note that the mammalian immune system must continuously deal with its own diverse microbiota, a huge external microbial load, and frequent microorganisms ingested by food and water.

## GUT MICROBIOTA AND DIETARY ASPECTS

The mammalian intestine contains a dynamic community of microorganisms that establish symbiotic relationships with their hosts, bringing essential contributions to human metabolic functions while living in a protected environment with conditions necessary for proliferation and obtaining nutrients. The intestinal microbiota is involved in the host's energy recovery from fermenting indigestible dietary substrates (*Yang et al., 2009*). Recent studies have demonstrated detailed insights into this mutually beneficial relationship. A germ-free murine model experiment showed that *Bacteroides thetaiotaomicron* induced expression of sodium/glucose transporters and absorption of dietary glucose released by bacterial digestion in the intestinal epithelium. The gut microbiota is rather beneficial for the absorption of calcium, magnesium, and iron, as well as the synthesis of vitamins, including biotin, folic acid, vitamin K, vitamin B12, and pantothenate; however, it may produce potentially toxic molecules, thus triggering DNA damage (*Rabizadeh & Sears, 2008*). Furthermore, each type of carbohydrate can affect the composition of bacteria in the intestinal microbiota, and the composition in turn can affect the metabolism of these molecules. Indeed, the composition of the microbial community can affect its ability to metabolize food carbohydrates. Vegetable starch, which is rich in amylopectin or amylose, is a common component in food and is metabolized by *Bifidobacterium*, *Bacteroides*,

and *Fusobacterium*. Diet can modulate the composition as well as the metabolism of the gut microbiota. High carbohydrate and fat intake can lead to increases in the populations of bifidobacterial and *Bacteroides* spp., respectively. Importantly, how diet affects colonial bacteria has been studied extensively. For example, carbohydrate-rich diets that predominate on the African continent favor Bacteroidetes, which can degrade xylan and cellulose to use energy from vegetable-based diets (*Wang et al., 1999*).

The Western diet, which is generally characterized by high intakes of fat and animal proteins, is often associated with the phylum Firmicutes, which alter the host's metabolic activity. A study comparing intestinal bacteria in infants (3 weeks to 10 months), adults (25–45 years), and elderly individuals (70–90 years) eating a western diet found dramatic differences among the three groups. There were significant differences in the average ratios of Firmicutes to Bacteroidetes in either infant or elderly group compared with adults (*Glendinning et al., 2014*).

Interactions between dietary lipids and the gut microbiota have also been studied extensively. Given that fatty acids can lyse and solubilize bacterial cell membranes, they are known to have a broad spectrum of antimicrobial activity. Lipids in diet affect not only antimicrobial activity but also ATP production in bacterial cells. Treatment targeting diseases related to dyslipidemia can alter intestinal bacteria. Therefore, it is recommended to supplement the diet with fibers to facilitate the growth and activity of prebiotics and probiotics (*Shilling et al., 2013*). Moreover, gut bacteria metabolize dietary proteins into amino acids, as well as for immunoprotection and signaling molecules. Bacterial proteinases and peptidases work together and can break larger molecules into fragments for better absorption and utilization. For example, L-histidine can be converted into amine and histamine by histidine decarboxylases, which are produced by intestinal bacteria. These bacterial-produced histidine carboxylases can suppress the production of proinflammatory TNF; thus, the use of specific probiotics can be a strategy to modulate and alleviate chronic diseases caused by immune deregulation (*Thomas et al., 2012*).

Viral infections are another major factor that can significantly modulate the composition of gut microbiota, especially in infants. *In vivo* models have well elucidated different responses to live-attenuated rotavirus vaccines, especially in children living in low- to high-income countries, depending on multiple factors such as lifestyle, malnutrition, zinc deficiency, avitaminoses, and gut commensals (*Desselberger, 2017*).

As mentioned above, there is a significant correlation between the host's dietary habitat and the intestinal microbiota, but medications can also influence the composition of the microbiota.

## ANTIMICROBIALS AND ANTHELMINTICS

The misuse of antibiotics without the correct guidance or their indiscriminate use leads to resistance and selection of microorganisms. Resistant strains are selected when the drug is not used according to its function and dose and time of use essential for successful therapy. With the increase in resistance by microorganisms, treatment of many diseases would be compromised, which could even lead to death, and greater public health expenditure

would be needed to try to solve the problems ranging from diagnosis to definitive cure (*Willing, Russell & Finlay, 2011*).

The most commonly described resistant bacteria are *Pseudomonas* spp., *Klebsiella* spp., *Acinetobacter* spp., *Escherichia coli*, and methicillin-resistant *Staphylococcus aureus* (MRSA) (*Silvestri, Lenhart & Fox, 2001*). The identification of drug-resistant microbes is a time-sensitive task; treatment for bacterial infections is assigned according to an established protocol and not with detailed bacterial identification. Drug susceptibility has been considered a factor that contributes to the increased mortality rates, especially in hospitalized patients (*World Health Organization, 2012*).

The treatment for helminths is based on albendazole and mebendazole, which bind to parasite β-tubulin and inhibit parasite microtubule polymerization, thereby causing the death of adult worms (*Bethony et al., 2006*). For the treatment of schistosomiasis, praziquantel (PZQ) is the drug of choice. In the lowest effective concentration, it causes increased muscle activity followed by contraction and spastic paralysis. At higher therapeutic concentrations, PZQ causes vacuolization and vesiculation of the tegument. This effect results in the release of the parasite content, activation of the host's defense mechanism, and destruction of the worms (*Siqueira et al., 2017*). Unfortunately, not only antibiotics have problems with resistance. Strains obtained from places where schistosomiasis is endemic show different sensitivity to PZQ, and this phenomenon could be related to the previous contact with the parasite; thus, in cases of reinfection, a different treatment is necessary (*Cioli et al., 2004*). The use of a unique drug for treatment has been studied over the years, and the results have shown the resistance of parasites (*Cioli et al., 2004*), which has reduced the percentage of cures in African countries, such as Senegal (with a cure rate of only 18%) and Kenya, indicating a substantial variation in drug efficacy in children (*King, Muchiri & Ouma, 2000*; *Gryseels et al., 2001*). Other therapies based on the interaction of microbiota and helminths are being used for treatment. For IBD and celiac disease, the use of antihelminth therapy is a favorable pathway because altering intestinal permeability and the host's immune response to a Th2 cytokine-mediated response can modulate the host defense (*Sipahi & Baptista, 2017*; *Vale et al., 2017*).

## MICROBIOTA AND PARASITIC DISEASES

The host's defense mechanisms against colonization by pathogens are related to habitat competition for nutrients and fixation sites, and the production of antimicrobial compounds and metabolites that may be unfavorable for parasites (*Hooper, Littman & Macpherson, 2012*; *Ogaki, Furlaneto & Maia, 2015*). Parasites attempt to modulate the host's immune system. The immune responses directed to bacteria and helminths are different, with effector mechanisms of helper T cells involving T helper 1 and Th17 for bacteria and Th2 for helminths. Some bacteria such as *Bacteroides fragilis* and *Clostridium* spp. can suppress the immune response by the induction of regulatory T cells (Tregs) (*Round & Mazmanian, 2010*; *Thomas et al., 2012*).

It has been reported that helminth infections modify the Th2 activity and damage the immune homeostasis, and they may be further involved in functional changes of intestinal

bacteria (*Ahmed et al., 2016*). Under the predominance of Th2 cells, cytokines are not sufficient to remove adult worms from the intestine. In addition, even with a long-lasting Th2 response, infected individuals show no signs of an evident worm allergy and may be protected against the development of allergies (*Loukas et al., 2016*; *Brosschot & Reynolds, 2018*). Helminths are still a huge problem for developing countries, and each helminth has particularities in interacting with the gut microbiota.

## ASCARIASIS

Ascariasis is a disease caused by a widely distributed geohelminth nematode *Ascaris lumbricoides*. It occurs mostly in tropical and subtropical countries as well as sporadically in the developed areas of the world (*Dold & Holland, 2011*). The parasite can infect reptiles, fish, birds, and mammalians, and the transmission occurs *via* water and food contaminated with eggs. *Ascaris lumbricoides* infects humans *via* fecal–oral transmission. In brief, passing through four developmental stages, L1 to L4, fertilized embryonated eggs become adult worms in the host's intestine. Importantly, 2 to 3 months after the infection, adult female worms produce thousands of eggs daily and pass them *via* stool. Adult worms in the host and eggs in moist warm soil can remain for years (*Nejsum et al., 2012*).

The host's microbiota provides the direct environment to *Ascaris*. However, Midha et al. have shown that products of nematodes (*e.g.*, excretory-secretory products (ESP), body fluids (BF)) extracted from intestine-dwelling life stages of *Ascaris suum* induce broad-spectrum antimicrobial activity upon immediate gut microbiota. Based on these findings, the gut microbial alterations may also depend on indirect changes in the host's immune system and metabolic activities influenced by nematode products (*Acevedo et al., 2011*; *Midha et al., 2018*).

The use of polyphenols is described in the literature, and this type of substance can modulate the inflammatory and immune responses of the mucosa and regulate the parasitic load. In a study with *A. suum*, polyphenols utilized in feed for pigs were helpful as they could modulate the responses against the parasite. The gene CCL26, which encodes a chemokine that regulates the recruitment of eosinophils, was downregulated when polyphenols were administered, showing that polyphenols can modulate the immune system (*Easton et al., 2019*).

Moreover, a study based on persistent depression showed that behavioral and host's physiological axis was altered by a complex network of communication among the host, microorganisms, and macroorganisms. It was also suggested how *A. lumbricoides* was associated with a subnetwork of the gut microbiota (*e.g.*, reduction in the number of species that compose each of the genera, interaction among other gut microorganisms) and induction of human depression. Taken together, *Ascaris* spp. can modulate the human gut microbiota for its own benefit (*Ramirez et al., 2020*).

## TRICHURIASIS

Trichuriasis is one of the most common soil-transmitted helminthic (STH) diseases. It is estimated that about 604–795 million people are infected all over the world (*Vos et al.,*

*2015*; *World Health Organization, 2020*). After about 12 weeks from ingestion of *Trichuris* eggs, the released larvae become adults within the colon (especially cecum and ascending colon) epithelium, where they burrow (*World Health Organization, 2012*). Increasing evidence shows the interactions between the host's microbiota and *Trichuris* infection. Specifically, gut bacteria play an important role in whipworm colonization, which triggers the development of the host's immune system (*Elliott, Mpairwe & Quigley, 2005*; *Cooper et al., 2013*).

*T. muris* (mouse whipworms) is an ideal model for studying *Trichuris* infection and associated pathophysiological changes (*White et al., 2018*; *Lawson, Roberts & Grencis, 2021*). A study based on this model has identified that *Trichuris* infection can modulate mouse intestinal microbiota; specifically, it reduces the abundance and the diversity of Bacteroidetes, including Parabacteroides and Prevotella (*Houlden et al., 2015*). *T. muris* infection in mice further alters metabolic products compared with the uninfected control group. There was a significant elevation in the number of essential amino acids (*e.g.*, phenylalanine and threonine), depletion of vitamin D2/D3 derivatives and glycerophospholipids, large quantity and range of fatty acids and intermediates involved in amino acid synthesis (*e.g.*, biosynthesis of phenylalanine, tryptophan, tyrosine), and breakdown products of plant-derived dietary carbohydrates. The remodeling of metabolic products in the infected mice reflects the ability of the mice to maximize nutrient release from their diet, which may involve the modulation of the intestinal microbiota. Interestingly, it was further revealed that there was a close connection between the chronic responses of the parasite-infected hosts and lasting immunological tolerance in mice with intestinal dysbiosis (*Elliott, Mpairwe & Quigley, 2005*). In contrast, there was no notable association between trichuriasis and the composition of the fecal microbiota of children compared with uninfected subjects. However, because of the limited sample size, this research group suggested further studies with heavily infected children and healthy individuals to replicate the same laboratory investigations and verify the effects of trichuriasis on the gut microbiome (*Cooper et al., 2013*).

According to *White et al. (2018)*, *T. muris* acquires its own microbiota to establish itself in the host's intestine, which depends on the existing microbiota in the host. In this way, the host's microbiota also changes, which implies a change in the host's health. In this study, the mice in the control group showed an equal predominance of Bacteroidetes and Firmicutes in the intestinal tract, whereas the helminth-infected mice group presented a decreased proportion of Bacteroidetes and an increased proportion of Firmicutes in the intestinal tract, and there was a significant reduction in the total bacterial species diversity. There are indications that Pseudomonadota would be more interesting for helminths, particularly because helminths inhabit regions with higher oxygenation, which are more suitable for Pseudomonadota. In terms of *Trichuris* microbiota, most of the identified bacteria belong to the Lachnospiraceae family and Bacteroidales subgroup. There are also indications that as the infection occurs, helminths adjust and modify the host's microbiota (*Elliott, Mpairwe & Quigley, 2005*; *Cooper et al., 2013*; *White et al., 2018*). Apparently, after the changes to microbiota, subsequent helminth infections are inhibited. This process in a way promotes a chronic infection.

Insufficient colonization of the gastrointestinal tract and/or respiratory tract by commensal microorganisms regulates the immune responses and may favor the development of atopy and asthma, which occurs in individuals of all ages, but frequently begins in childhood (*Frati et al., 2018*). In addition, *Rodrigues et al. (2008)* reported that children with heavy infection of *T. trichiura* in early childhood have a drastic reduction in the development of asthmatic reactions in their later childhood, even in the absence of *T. trichiura* infection in later childhood. Taken together, *T. trichiura* infection may influence the immune system of children immensely and reduce the allergen skin test reactivity significantly.

In a recent review (*Lawson, Roberts & Grencis, 2021*), it has been reported how the gut microbiota could influence the immune response. The intestinal microbiota contributes to the establishment of human health by acting in nutrition, the control of pathogens, and the development of the immune response. The microbiota, in turn, can be modified by diet and with the use of antibiotics, which could bring eventual consequences for human health. The review has also highlighted related different animal models with *T. muris,* showing evidence that a helminth depends on the microbiota to establish itself in the intestine, while antibiotic treatment can interfere with this process. In general, different *in vivo* models may help to understand how trichuriasis, the microbiota, and the host's immune response interact with each other, which may be helpful to advance the treatment of autoimmune diseases using helminth antigens such as those against *Trichuris*.

## ANCYLOSTOMIASIS

Ancylostomiasis is also known as hookworm infection. It is mostly caused by *Ancylostoma duodenale* and *Necator americanus.* It is widely spread in poor socioeconomic countries in tropical and subtropical areas, and the global prevalence of any hookworm infection is estimated at around 576–740 million cases (*Stracke, Jex & Traub, 2020*; *Centers for Disease Control and Prevention, 2020*).

Recently, a low dose of hookworm administration has been used as a therapeutic intervention for certain human diseases (*e.g.*, Crohn's disease); however, there are a limited number of studies to elucidate the influence of the administered parasite on the human gut microbiota (*Loukas et al., 2016*; *Idris, Wintola & Afolayan, 2019*). *Cantacessi et al. (2014)* found that experimental administration of *N. americanus* to healthy individuals did not affect fecal microbiota, but they did not reject the possibility of having minor changes to microbiota at the site of infection. *Ducarmon et al. (2020)* showed an increase in the species richness of the gut microbiota among all volunteers during an established infection, but the diversity and stability were almost unchanged. Even the group with many symptoms was characterized by transient microbiota instability and subsequent recovery. *Barnesiella, Lachnospiraceae, Bilophila,* and *Escherichia–Shigella* were the most often encountered genera, but *Allisonella* was the most encountered genus in individuals with few symptoms. Hence, individuals with the more unstable microbiota after infection are more likely to experience gastrointestinal symptoms during infection, or gastrointestinal symptoms could be caused by more severe enteritis that also affects the microbiota stability. *N.*

*americanus* may play a pivotal role in the upregulation of anti-inflammatory cytokines, such as interleukin, which may further shape the host's immune response (*Hooper, Littman & Macpherson, 2012*; *Loukas et al., 2016*; *Hailegebriel, Nibret & Munshea, 2020*). Regarding the interaction with microbiota during the acute phase of ancylostomiasis in humans, it was demonstrated that there were no differences in microbiota in feces relative to healthy individuals; however, a biopsy of the places where the parasites are fixed could reveal some differences. Another important factor is the intensity of the infection, as it was shown that alterations in microbiota were observed with a higher number of parasites (*Loukas et al., 2016*). Owing to practical and ethical issues, the composition of the gut microbiota is primarily measured by fecal microbial analysis. However, one research group analyzed duodenal biopsy tissues taken from individuals with celiac disease (CeD) after an experimental hookworm infection along with gluten ingestion. Providing supportive insights from previous studies and their observations, the group reported qualitative and quantitative changes to the gut microbiota, especially at the site of infection in individuals with active CeD. In a similar group of subjects infected with experimental hookworm along with dietary gluten, there was an association between the intestinal tissue–resident microbial species richness (many species within the Bacteroides phylum) and their diversity (*Giacomin et al., 2016*). Interestingly, when analyzing the alteration of microbiota and cognition effects, it was found that perception was altered and the microbiota diversity was reduced in hamsters infected with hookworms. In many situations, these animals had poorer performance when compared with noninfected animals, and the lack of conventional behavior, which included recognition of places, objects, and preferences determined after an encounter, was compared with the noninfected animals, proving that helminths have a direct impact on brain health (*Pan et al., 2019*).

In a study comparing the treatment and placebo in humans, it was found that the alpha-diversity was elevated in the group infected with *N. americanus*, but not in the placebo group. These effects could be related to alleviation of allergic or immunological disarrangement (*Jenkins et al., 2021*).

## SCHISTOSOMIASIS

Schistosomiasis, also known as bilharzia, is an infectious tropical parasitic disease caused by a group of blood flukes called schistosomes. Over 230 to 250 million people are annually infected with schistosomes, and nearly 700 million people are at risk of infection in endemic areas (*Colley et al., 2014*; *Nelwan, 2019*). Schistosomes have a complex life cycle, which involves humans (mammalian), snails, and freshwater. There are three main species of schistosomes (*Schistosoma haematobium, S. mansoni,* and *S. japonicum*) that cause human disease according to parasite distribution, the kind of disease that they cause, and the type of snail involved in the parasitic life cycle. Adult male and female schistosomes can be found within the veins of their human host (*Colley et al., 2014*). Normally, schistosomula, a stage of the parasitic life cycle, migrate through the blood and lymphatic system to the lungs and then the liver, where the parasites become mature and fertile. However, the final destinations of these parasites are either perivascular or mesenteric venules. Adult male

and female worms that reach mesenteric veins of the lower plexus of the large intestine in humans mate and shed fertilized eggs through feces or urine. The retained eggs in nearby host tissues can induce various local and systemic pathophysiological changes (*e.g.*, impaired cognition, anemia, growth stunting, periportal fibrosis with portal hypertension, scarring, urogenital inflammation) *via* immune-mediated (CD4+ T-cell dependent) granulomatous responses (*Colley et al., 2014*; *Holzscheiter et al., 2014*).

Schistosoma spp. cause several pathological processes in the intestinal epithelium that, associated with the immunomodulatory response, can lead to a decrease in the protective barrier against bacteria (*Barnhill et al., 2011*). Thus, these conditions favor the translocation of the intestinal lumen bacteria into the bloodstream. The use of PZQ, an anthelmintic, changes the composition of the microbiota irreversibly, suggesting that exposure to such treatment in early childhood may have long-term negative health impacts due to the alterations to the community of beneficial microorganisms (*Schneeberger et al., 2018*).

Using a murine model, *Holzscheiter et al. (2014)* have shown that oral administration of broad-spectrum antibiotics and antimycotics significantly reduces both gut microbiota and inflammation, resulting in less granuloma development. Moreover, they noted skewed schistosome-mediated immune markers, suggesting that the host microbiota acts as an intermediate (third partner) to initiate schistosome-specific immune responses and further reduce gut microbiota pathological changes (*Holzscheiter et al., 2014*). Another study demonstrated that the host's commensal bacteria during infection by *S. mansoni* played an important role in the formation of intestinal granulomas and specific immune responses of schistosomiasis, which may influence the excretion of eggs (*Barnhill et al., 2011*). There are a few case reports related to septicemia, which demonstrated the coinfection with gut bacteria and *S. mansoni* (*Muniz-Junqueira, Tosta & Prata, 2009*; *Barnhill et al., 2011*; *Hsiao et al., 2016*). Once in circulation, these gut bacteria reach the adult worms of *S. mansoni* present in mesenteric veins and colonize the cecum of the parasite (*Muniz-Junqueira, Tosta & Prata, 2009*). The association of bacteria with *S. mansoni* enables prolonged bacterial infections, the development of antibiotics resistance, and the ineffective treatment of both infections (*Barnhill et al., 2011*). *Cortés et al. (2020)* analyzed schistosomiasis and intestinal microbiota modulation by comparing human microbiota-associated mice (HMA) and wild-type mice (WT) both infected with schistosomes. In the gut microbiota of WT animals, similar proportions of Bacteroidetes and Firmicutes were observed. In contrast, the phylum Bacteroidetes predominated in HMA mice. Phylum Proteobacteria is considered a marker of dysbiosis in the intestinal microbiome, as well as a critical determinant of the host's health, metabolism, and inflammation (*Shin, Whon & Bae, 2015*).

S. haematobium infection decreases the abundance of phylum Firmicutes and increases the prevalence of phylum Proteobacteria (*Ajibola et al., 2019*). In a clinical study, an altered gut microbial composition was noted when comparing schistosome-infected ($n = 24$) and noninfected ($n = 25$) groups of adolescents. Hence, as many individuals are infected by schistosomes, their intestinal microbiota may also be altered. Interestingly, the same was found when analyzing *S. japonicum* in a population of 11 patients infected in China; Firmicutes and Bacteroidetes were the most altered phyla as observed for *S. mansoni* and *S. haematobium*. *Bacteroides*, *Faecalibacterium*, and *Prevotella* were the most often

found genera, which could be related to difficulties in the metabolism of many nutrients; hence, a problem observed in patients with schistosomiasis with intense infections can be related to this dysbiosis (*Jiang et al., 2021*). After the infection with *S. japonicum*, Bacilli and Lactobacillales showed an increase, and this alteration was related to cirrhosis (*Gui et al., 2021*). Therefore, administration of PZQ can be helpful to elucidate how long patients must be treated to eliminate parasites and their eggs from the host and reestablish the altered intestinal microbiota. *Schneeberger et al. (2018)* showed that the compositions of phylum Firmicutes and Proteobacteria were influenced by schistosomiasis, and both were dominant, whereas the different observations upon schistosomiasis and PZQ anthelmintic administration could depend on the study population and the quality of life. This may imply the influence of the places of residence of the population, where this one was in a rural area of Africa and not in large cities.

Moreover, the regulatory functions of the gut microbiota in response to *S. japonicum* infection in mice were studied (*Zhang et al., 2020*). *S. japonicum* infection reduced the gut microbiota and reduced granuloma formation and fibrosis. It was possible to partially reverse the aforementioned intestinal microbial changes by cohousing *S. japonicum*-infected mice with noninfected controls, resulting in lower intestinal pathological responses. Furthermore, the authors reported reduced levels of Th1-associated IL-4, IL-5, and IL-13, and increased levels of IL-10 and TGF-β (limiting excessive Th1 and Th2 immune response) in mice infected with *S. japonicum* and depleted gut microbiome, and suggested that the regulatory functions of the intestinal microbiota in *S. japonicum*-infected mice were mediated by alteration of local immune responses.

*Hu et al. (2020)* also reported changes in the gut microbiome (decreased abundance and diversity of the flora) in infected animals compared with controls as the disease progressed from acute to the chronic phase. Changes in AMP-activated protein kinase and chemokine signaling pathways were also observed after infection, as analyzed by metagenomic analysis. In parallel, the metabolic profile of these animals was evaluated, where some markers of the initial infection were identified, such as phosphatidylcholine and colfosceril palmitate in the serum, and xanthurenic acid, naphthalenesulfonic acid, and pimelylcarnitine in the urine.

## CONCLUSION

In this study, it was found that the interaction between microbiota and parasites is complex and needs attention. *A. lumbricoides, T. trichiura, N. americanus, A. duodenale,* and *S. mansoni* have different forms of interaction with the host's microbiota, but share some mechanisms of activation of immune defense. While most of them (*e.g.*, *Ascaris* sp. and *Trichuris* sp.) can modify the gut microbiota, others (such as *N. americanus*) appear not to influence it. Although new studies have been done, not all mechanisms have been elucidated, such as how it is possible to manipulate the microbiota beneficially for the host; this is because most of these studies were done in a controlled infection, which is quite different from the real world where a person can have a huge infection or coinfection. Moreover, given that anthelminthic drugs can change the composition of the microbiota

in *Schistosoma* infection, it is worth further studying whether depletion of certain bacteria could minimize the effects of an infection. It is conceivable that the presence of specific bacteria in the gut microbiota could protect the host against infection by intestinal helminths, but it is necessary to reach a complete understanding of the interaction between the host, bacteria, and parasites to develop new treatments.

## ACKNOWLEDGEMENTS

We are thankful to the staff of Dokkyo Medical University, Universidade Federal de Alfenas Niigata University, and The Open University Sri Lanka for their logistical support. We thank Mr. Clyde Ito, Department of Pediatrics, Dokkyo Medical University for reading and improving the manuscript.

### Funding

This study was supported by the Coordenação de Aperfeiçoamento de Pessoal de Nível Superior - Brasil (CAPES) - Finance Code 001 and JSPS KAKENHI Grant Number 21K12269. The funders had no role in study design, data collection and analysis, decision to publish, or preparation of the manuscript.

### Grant Disclosures

The following grant information was disclosed by the authors:
The Coordenação de Aperfeiçoamento de Pessoal de Nível Superior - Brasil (CAPES) - Finance Code 001 and JSPS KAKENHI: 21K12269.

### Competing Interests

The authors declare there are no competing interests.

### Author Contributions

- Matheus Pereira de Araújo conceived and designed the experiments, performed the experiments, analyzed the data, prepared figures and/or tables, authored or reviewed drafts of the paper, and approved the final draft.
- Marcello Otake Sato and Megumi Sato conceived and designed the experiments, analyzed the data, authored or reviewed drafts of the paper, and approved the final draft.
- Kasun M. Bandara WM and Luiz Felipe Leomil Coelho performed the experiments, analyzed the data, authored or reviewed drafts of the paper, and approved the final draft.
- Raquel Lopes Martins Souza, Satoru Kawai and Marcos José Marques analyzed the data, authored or reviewed drafts of the paper, and approved the final draft.

### Data Availability

 All data analyzed in this study are included in this published article.

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
