# Peer review of "Unbalanced relationships: insights into the interaction between gut microbiota, geohelminths, and schistosomiasis"

_PeerJ, doi:10.7717/peerj.13401_

## Round 0.1 · original submission · Major Revisions

The reviewers and I considered the topic an important issue for the specific field of knowledge. However, there are many flaws in the submitted manuscript.

The review requires more information about the methodology used to select the literature review; please, provide information about papers' inclusion and exclusion criteria.

Note that reviewer #3 mentioned other relevant papers that must be included in the review.

Another point that deserves attention refers to the necessity of critical and insightful analysis regarding the discussion of the analyzed papers.

The conclusion must be improved and deeper (see reviewer opinion #3).

The English language also needs to be improved to make the manuscript clear. Many parts of the text are hard to follow.

·

Basic reporting

The article presents an important review on the interaction of macrobiota (helminths) and microbiota (bacteria). The paper structure is very clear and easy to read.

Experimental design

Did the authors not use a temporary limit of consulted articles? If the authors used a specific literature review period, add in the section Survey methodology.

Validity of the findings

no comment

Additional comments

Just a few suggestions:
Add italics to species names (taxonomy rules) and in other cases, lines: 21, 137, 191, 226 (in vivo), 332 (Trichuris), 364-365
Line 290: It is a topic discussed by different research groups, based on the agreement or disagreement of the Hygiene theory. I suggest a small adjustment in the sentence: ...allergy and are actually protected..., change to: ...and may be protected against...
Some typos, lines: 247, 266, 288, 303 (worms), 311, 317, 426
Line 333: Trichuria colonization, change to whipworm colonization.
Lines 337, 370 : microflora, change to microbiota
Lines 356, 357 and 358: T. trichiuria, change to T. trichiura
Line 365: infestation, change to infection.
Line 366: review the number 450.68 million.
Line 469: disbiose, change to dysbiosis.

Reviewer 2 ·

Basic reporting

no comment

Experimental design

no comment'

Validity of the findings

no comment'

Additional comments

This manuscript is scientifically sound the results are important in the field and relevant this work contains important obserations about the regulatory capaity of parasites in the intestinal ecosystem. I suggest to publish it

Reviewer 3 ·

Basic reporting

This review is a timely and an important topic in regards to tropical medicine.
Major improvements in grammar and language are needed throughout, and in parts the text is very difficult to follow.
Line 35, the term depleted infers the whole (entire) resource has been used, reduced of diminished may be better.
Line 87-88, the authors are referring to Schistosoma but Ducarmon references is related to Necator. I would recommend the referencing here and in other parts of the paper with the recent work of Cantacessi for S. mansoni and Gordon for S. japonicum.
Other publications that should be considered include, PMCID: PMC7580221; PMC8660641 & PMC5851687 (Trichuris)
Table 1 is very useful.
Lines 252-3 the statement on PZQ is misleading since the drug has varying efficacy between developing and adult stages within the mammalian host.
Please be consistent with the use of abbreviations ie line 255 praziquantel should be PZQ.
Some sections contain material that is a little ad hoc ie lines 257 261 following on from PZQ and under the section title that doesn’t fit, then comes back to PZQ. Ideas sometimes do not flow particularly well, with some topics not connecting within paragraphs. Statement in lines 269-273 not supported by the text and seems not very feasible, considering novel antibiotics are very difficult to develop and their use to treat helminths would not be very appropriate.
Line 277, some reports present little to know change with helminth infections. Lines 277-281 is confusing.
Line 281 change deceive to modulate.
Line 284 suppress what?
Line 305, less concern or less information? This contradicts the statement in lines 323-324.
The breaking of sections into diseases, works particularly well.
Line 415 change form to stage.
Line 426 yet another referencing formatting error. Species names should be italicised in the reference list.
Line 445-6 “In another study…”, which one? No reference provided?
Lines 452-455 is that a clinical or animal study.
Lines 455-472, much of this is confusing. Line 474 “….were also studied.” Add reference here.
The conclusion section does not say too much and should instead state more specific approaches that should be taken and why.
trichiura is misspelt throughout

Experimental design

see above

Validity of the findings

see above

Additional comments

see above

---

## Round 0.2 · accepted · Accept

The authors addressed the main concerns raised. The manuscript is much improved by the modifications.